# TPR-Attention for Combinatorial Generalization

**Melisa Civelekoğlu & Isabeau Prémont-Schwarz**
Université Laval
Institut intelligence et données
Mila – Québec Artificial Intelligence Institute
`melisa.civelekoglu.1@ulaval.ca, isabeau.premont-schwarz@ift.ulaval.ca`

## Abstract

Systematic generalization remains a significant challenge in deep learning. In particular, **combinatorial generalization** – generalizing to new configurations of known factors of variation – is effortless for humans but difficult for standard neural architectures that rely on statistical correlations rather than explicit structural representations. We introduce a new architectural component that embeds structured inductive bias into deep learning: an attention mechanism operating over **tensor-product representations (TPRs)**. Through controlled experiments on compositional tasks, we show that this TPR-Attention mechanism outperforms existing architectural components in combinatorial generalization. These results highlight the value of integrating explicit compositional structure into neural attention and point toward a promising path for models capable of systematic generalization.

## 1 Introduction

A natural ability of human intelligence is the recombination of known factors of variation into novel structures, known as **combinatorial generalization** (Cole et al., 2013; Smolensky et al., 2022). Rather than memorizing correlations, it relies on the compositionality of many real-world decision-making problems. For example, a human who has learned the meanings of "throw", "catch" and "throw fast" could effortlessly understand the meaning of "catch fast".

Traditional deep networks, on the other hand, struggle with such seemingly straightforward combinations (Lake & Baroni, 2018) as they rely on statistical patterns rather than compositional rules. A model trained on red circles and green squares, for example, should also be able to generate red squares and green circles. Despite remarkable successes, deep networks remain limited because they learn statistical co-occurrences rather than underlying compositional structure and thus are still far from achieving human-level generalization.

To address this limitation, methods for encoding symbolic structure, such as Tensor Product Representations (TPRs) (Smolensky, 1990), have been proposed. In this work, we make two contributions. First, we introduce a new neural architectural component, **TPR-Attention**, which uses a structured object-centric representation built from the binding of an object's factors of variation, and whose attention mechanism explicitly performs binding and unbinding operations on these objects. Second, we provide empirical evidence that TPR-Attention achieves better combinatorial generalization than existing architectural components, particularly in the difficult setting where factors of variation interact(Montero et al., 2024).

## 2 Related Work

Many previous works have focused on learning disentangled representations (Mathieu et al., 2019; Wang et al., 2024), i.e., independent factors of variation that can be recombined to

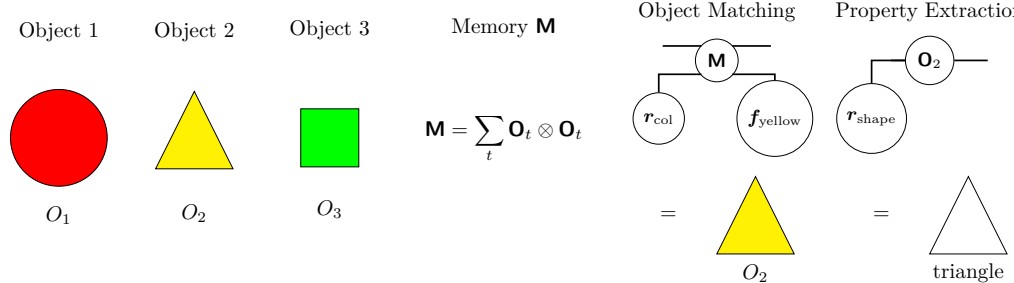

Figure 1: Illustration of the object matching and property extraction phases of TPR-Attention: Objects $O_1$, $O_2$, and $O_3$ are stored in memory $\mathbf{M} = \sum_t \mathbf{O}_t \otimes \mathbf{O}_t$. A query for a filler match $\boldsymbol{f}_m = \boldsymbol{f}_{\text{yellow}}$ and role match $\boldsymbol{r}_m = \boldsymbol{r}_{\text{col}}$ retrieves the yellow triangle $O_2$. Finally, a query for a target role $\boldsymbol{r}_t = \boldsymbol{r}_{\text{shape}}$ retrieves the triangle shape from the matched $O_2$.

produce novel concept compositions. One motivation for disentanglement is that separating these factors may improve generalization, either by capturing inherent compositional structure (Duan et al., 2020) or by isolating causal variables (Schölkopf et al., 2021).

However, the relationship between disentanglement and compositional generalization remains unclear. Several studies report limited evidence that explicit factor decoupling improves generalization (Xu et al., 2022; Montero et al., 2024). In particular, Montero et al. (2024) show that models can achieve high disentanglement by mapping perceptual inputs to factors that remain invariant across examples, yet still fail to generalize compositionally when factors interact. In this paper, we focus specifically on this hard case of *interacting* factors of variation, where existing approaches are known to struggle.

There exist explicit symbolic structures such as TPRs and other Vector Symbolic Architectures (Kleyko et al., 2022) which provide role–filler bindings that separate feature from content. In this paper, we provide transformation operations to be used between these structures with a focus on combinatorial generalization.

## 3 METHOD

### 3.1 REPRESENTATION SPACE

Objects are represented by TPRs, which provide a vector-space embedding of symbolic structures. Each object is composed of roles, which specify an attribute slot of an object (e.g., *shape*, *colour*, *position*), and fillers, which are the values occupying each role (e.g., *square*, *red*, *center*).

We represent each role by a role vector $\boldsymbol{r}$ and each filler by a filler vector $\boldsymbol{f}$. An object is defined as the tensor $\mathbf{O}$, where the tensor product is used as the binding operator, so that a role-filler relation is described by $\boldsymbol{r} \otimes \boldsymbol{f}$. We assume orthogonality of role vectors, i.e., $\boldsymbol{r}_i^\top \boldsymbol{r}_j = \delta_{ij}$, to prevent interference across fillers.

The final TPR form of an object is the superposition of all its role–filler bindings:

$$\mathbf{O}_i = \sum_j \boldsymbol{r}_j \otimes \boldsymbol{f}_j^i.$$

### 3.2 TPR-ATTENTION MECHANISM

There are three stages in TPR-Attention: (i) generate a structured role-filler query and match relevant objects, (ii) extract a target property from each matching object and (iii) transform each extracted property and re-bind it to a new role per attention head.

To match and extract at the same index $t$, we define a **TPR-based structured associative memory mechanism (TPR-SAM)** (see Appendix C for a generalization):

$$\text{TPR-SAM}(\mathbf{O}_t) := \mathbf{O}_t \otimes \mathbf{O}_t.$$

Then, TPR-Attention mechanism operates on a TPR-SAM memory, represented as the superposition of all past transformed objects:

$$\mathbf{M}_t \leftarrow \mathbf{M}_{t-1} + \mathbf{O}_t \otimes \mathbf{O}_t = \sum_{s=1}^{t} \mathbf{O}_s \otimes \mathbf{O}_s.$$

### 3.2.1 OBJECT MATCHING

Given a role $\boldsymbol{r}_m$ and filler $\boldsymbol{f}_m$ to match, each object is weighted by a similarity score between $\boldsymbol{f}_m$ and the filler bound to $\boldsymbol{r}_m$ (see Appendix A for notation and B.1 for a detailed derivation).

$$\text{M}_{\text{obj}}(\mathbf{M}, \boldsymbol{r}_m, \boldsymbol{f}_m) := \quad \text{[diagram]} \quad = \sum_t \text{[diagram]} \quad = \sum_t (\boldsymbol{r}_m^\top \mathbf{O}_t \boldsymbol{f}_m)\, \mathbf{O}_t.$$

### 3.2.2 PROPERTY EXTRACTION

Given a target role $\boldsymbol{r}_t$, target fillers are extracted from the matching object and weighted by their similarity to the filler bound to $\boldsymbol{r}_t$ (see Appendix B.2).

$$\text{E}_{\text{prop}}(\mathbf{O}, \boldsymbol{r}_t) := \quad \text{[diagram]} \quad = \boldsymbol{r}_t^\top \mathbf{O}.$$

### 3.2.3 TRANSFORMATION AND RE-BINDING

Finally, the extracted fillers will be transformed by some learned linear transformation $\boldsymbol{H}$ and re-bound to a new role $r_n$ per attention head (see Appendix B.3). This allows multiple features to be extracted and transformed in parallel. The final output object is a superposition of these role-filler pairs per head.

$$\text{T}_{\text{bind}}(\boldsymbol{f}, \boldsymbol{H}, \boldsymbol{r}_n) := \quad \text{[diagram]} \quad = \boldsymbol{r}_n \otimes (\boldsymbol{f}^\top \boldsymbol{H}).$$

Finally, the full TPR-Attention mechanism per head is described by:

$$\text{TPR-Attn} = \text{T}_{\text{bind}}\left(\text{E}_{\text{prop}}(\text{M}_{\text{obj}}(\mathbf{M}, \boldsymbol{r}_m, \boldsymbol{f}_m), \boldsymbol{r}_t), \boldsymbol{H}, \boldsymbol{r}_n\right) = \quad \text{[diagram]}$$

## 4 EXPERIMENTS

We evaluate our mechanism on a controlled composition task (see Figure 3), implemented through a feature substitution, adapted from Montero et al. (2024) using the dSprites dataset (Matthey et al., 2017). In our current setup, rather than working with sprite-level images, our mechanism operates on latent representations (see Appendix E for TPR construction

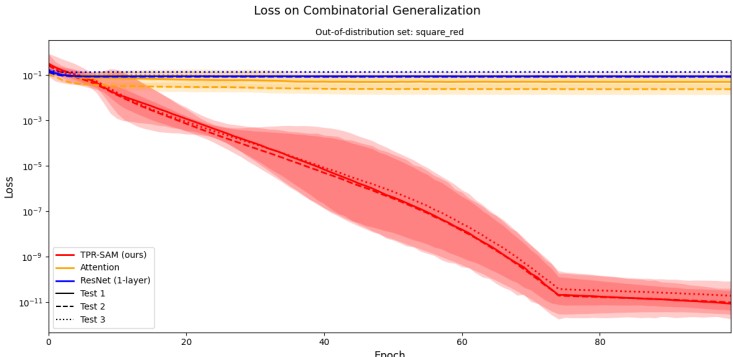

Figure 2: Loss on combinatorial generalization for the out-of-distribution set square_red. TPR-Attention with 4 heads (red) achieves lower loss in all tests compared to classical attention with 4 heads (yellow) and ResNet baseline with 1 layer (blue). Curves show the mean over 5 seeds, with shaded regions indicating ± 2 standard deviation.

details of the task), allowing us to test the composition mechanism on a controlled setting, independent of perceptual representation learning.

For each experiment, we compare one layer of TPR-Attention with a baseline of one layer of regular attention, and ResNet to see how well each generalizes to unseen combinations of factors of variation.

We evaluate generalization using three out-of-distribution (OOD) splits: *scale_pos*, *square_pos* and *square_red*. These sets are chosen to systematically evaluate compositional generalization under different combinations of **numerical** vs. **categorical** factors of variation. The *scale_pos* split tests the holdout of numerical features, where latents with scale $> 0.7$ and $posX > 0$ are held out during training. The *square_pos* split introduces a mixture of numerical and categorical features, holding out squares at $posX > 0$. Finally, to evaluate over categorical features, we additionally assign a random color encoding (red, green, or blue) to each latent input and choose the *square_red* split, where red squares are held out.

We further evaluate our TPR-Attention on three different test conditions: (Test 1) only the reference input contains OOD factors, (Test 2) only the transform input contains OOD factors, and (Test 3) both the reference and the transform input are in-distribution, yet their composition yields an OOD composition in the output.

Finally, we evaluate two different experimental settings: **non-interacting** and **interacting** factors of variation. This is motivated by prior work which has shown that compositional generalization becomes substantially more difficult when factors interact (Montero et al., 2024). In the non-interacting setting, factors of variation are independent, such that substituting one factor does not affect another. For example, changing an object's color does not affect its shape. In contrast, in the interacting setting, we add an extra component in the representation which is the interaction of two of the factors of variation. We consider two types of factor interac-

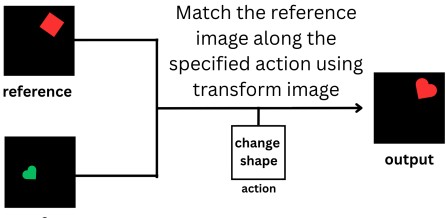

Figure 3: Controlled composition task. Given a reference input, transform input and an action corresponding to a factor of variation, the output must match the transform input along the specified factor and match the reference input for the rest of the factors. In the current experiments, we operate on pre-encoded latent representations rather than raw pixel inputs to isolate the behaviour of TPR-Attention independent of perceptual representation learning.

tions. First, we evaluate interactions between *numerical* factors, *scale_pos*. To induce the interaction, we construct an additional latent factor by adding the scale and position values.

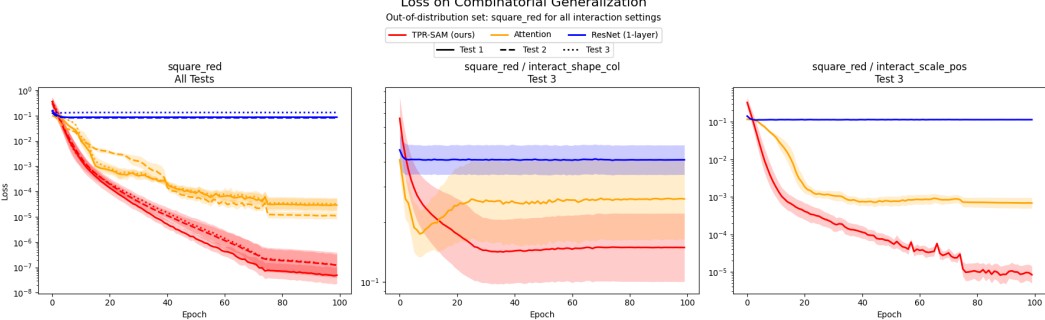

Figure 4: Loss on combinatorial generalization under different interaction settings for *square_red* OOD condition. TPR-Attention (red) is compared against classical attention (yellow) and a ResNet baseline with one layer (blue). Curves show the mean over five seeds, with shaded regions indicating ± one standard error. Refer to Appendix F for additional results.

Second, we evaluate interactions between *categorical* factors, *shape_col*. The interaction is introduced as a linear mixing of shape and color, mapping their representations with a random matrix $M$ (for further information, refer to Appendix E.3).

To operate on this composition task, we formulate an **action-conditioned TPR-Attention** model (described in Appendix D), where the input action determines a query $q_i$ that specifies the role-filler pair to be modified in the current object. We enable a copy mechanism, which initializes the output object with a copy of the reference input before applying action-conditioned updates. This design choice reflects the structure of the task, as the output must preserve most features of the reference object. To ensure a fair comparison, we implement analogous copy behavior in all baselines. In particular, ResNet baseline has a residual update that adds to the reference object and classical multi-head attention baseline attends over the reference and transform inputs, producing an update to be added to the reference object.

## 5   DISCUSSION

In this work, we introduced a new attention mechanism for structured TPR representations. This mechanism operates directly on role-filler bindings, giving the model explicit access to object-centric structure and enabling binding and unbinding operations within attention.

We then evaluated TPR-Attention in a controlled setting where the latent factors of variation interact. In this hard regime – where existing methods are known to struggle – TPR-Attention generalized compositionally out of distribution substantially better than classical attention and ResNets. To our knowledge, this level of combinatorial generalization on interacting factors has not been demonstrated by prior architectures.

A natural next step is to combine TPR-Attention with a learned encoder to operate directly on pixel inputs. Prior work (Mathieu et al., 2019; Wang et al., 2024; Xu et al., 2022; Montero et al., 2024) has shown that disentangled factors of variation can be recovered from images, suggesting that the structured representations required by TPR-Attention can, in principle, be learned rather than manually provided.

**Limitations.** Our experiments rely on manually provided latent factors, which allows us to isolate the behavior of TPR-Attention but does not yet demonstrate performance on raw perceptual inputs. In addition, our evaluation focuses on a controlled setting with a small number of structured factors, and it remains to be seen how TPR-Attention scales to higher-dimensional or noisier domains. Finally, another major limitation of the current work is that we only compared combinatorial generalization for a single layer. There is no guarantee that the generalization advantage will persist when we compare stacked versions of the base layer.

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

## A Tensor Network Diagram Notation

Tensors are multidimensional arrays which encode interactions across multiple modes. Vectors and matrices are a special case of tensors, which store information along one and two modes respectively.

We use tensor network diagrams as a graphical notation, where each node represents a tensor and each edge represents one tensor mode of the node it belongs to.

For example, a vector $\boldsymbol{v}$, a matrix $\boldsymbol{M}$ and a third-order tensor $\mathsf{T}$ would be represented, respectively, by the following diagrams:

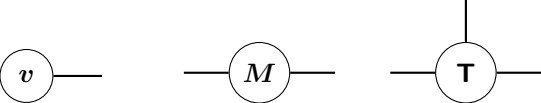

A tensor contraction is represented by a shared edge between two nodes corresponding to the mode along which contraction occurs. Edges that do not connect to another node correspond to free indices and represent the modes in the output of the operation. Examples are shown below.

**Example 1: Matrix-Vector Contraction.**

$$i \;-\!\!\!\!\left(M\right)\!\!\!\!\overset{j}{-}\!\!\!\!\left(v\right) \qquad = \qquad \sum_j M_{ij} v_j \qquad = \qquad i \;-\!\!\!\!\left(u\right)$$

**Example 2: Matrix-Tensor Contraction.**

$$i \;-\!\!\!\!\left(M\right)\!\!\!\!\overset{j}{-}\!\!\!\!\left(N\right)\!\!\!-k \qquad = \qquad \sum_j M_{ij} N_{jkl} \qquad = \qquad i \;-\!\!\!\!\left(O\right)\!\!\!-k$$

**Example 3: Tensor-Tensor Contraction.**

$$i \;-\!\!\!\!\left(A\right)\!\!\!\!\overset{k}{-}\!\!\!\!\left(B\right)\!\!\!-l \qquad = \qquad \sum_k A_{ijk} B_{klm} \qquad = \qquad i \;-\!\!\!\!\left(C\right)\!\!\!-j$$

Finally, tensor product (or outer product), is represented by placing tensors next to each other without shared edges.

**Example: Outer Product of Two Vectors**

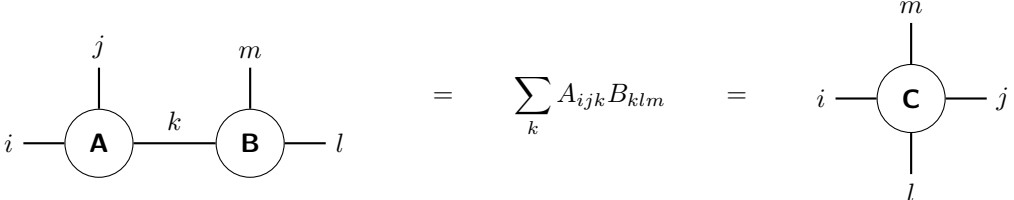

$$= \quad (\boldsymbol{v} \otimes \boldsymbol{u})_{ij} = v_i u_j$$

## B TPR-Attention Module

There are three stages in TPR-Attention: (i) object matching (see B.1), (ii) property extraction (see B.2) and (iii) transformation and re-binding per attention head (see B.3). The full computation of TPR-Attention is summarized in Algorithm 1.

---

**Algorithm 1** Full TPR-Attention

---

**Input:** Current object $\mathbf{O}_t$, memory of past objects $\mathbf{M}_t$
**Output:** Updated object $\mathbf{O}_{t+1}$
$\mathbf{M}_{t+1} \leftarrow \mathbf{M}_t + \text{TPR-SAM}(\mathbf{O}_t)$
**for** each head $n$ **do**
    Compute $\boldsymbol{f}_m^{(n)}, \boldsymbol{r}_m^{(n)}, \boldsymbol{r}_t^{(n)}$ as functions of $\mathbf{O}_t$
    $\mathbf{O}_{t+1,n} \leftarrow \text{T}_{\text{bind}}\Big(\text{E}_{\text{prop}}\Big(\text{M}_{\text{obj}}(\mathbf{M}_{t+1}, \boldsymbol{r}_m^{(n)}, \boldsymbol{f}_m^{(n)}), \boldsymbol{r}_t^{(n)}\Big), \boldsymbol{H}^{(n)}, \boldsymbol{r}_n\Big)$
**end for**
$\mathbf{O}_{t+1} \leftarrow \sum_n \mathbf{O}_{t+1,n}$

---

## B.1 OBJECT MATCHING

The following is the derivation of object matching in TPR-Attention:

First, we expand each $\mathbf{O}_t$ into its role-filler decomposition. Then, using the orthonormal property of the role vectors (i.e., $\langle \boldsymbol{r}_m, \boldsymbol{r}_j \rangle = \delta_{jm}$), we collapse the role tensor contractions via the Kronecker delta. The Kronecker delta eliminates the sum over $j$, leaving only the terms where $j = m$.

Finally, for each object $s$, the contraction produces a similarity score between $\boldsymbol{f}_m$ and $\boldsymbol{f}_m^s$. The final result is then a weighted sum of objects from memory, each scaled by how well $\boldsymbol{f}_m$ matched $\boldsymbol{f}_m^s$.

## B.2 PROPERTY EXTRACTION

In order to retrieve a target property, we introduce a target role $\boldsymbol{r}_t$ to extract target fillers from the retrieved objects and define:

As before, we use Kronecker delta to eliminate the sum over $i$, leaving only the terms where $i = t$.

### B.3 Transformation and Re-binding in Multi-Head TPR-Attention

Finally, the weighted sum of target fillers across retrieved objects are transformed by a learned matrix $\boldsymbol{H}$ and re-bound to a new role $\boldsymbol{r}_n$ per $n$ attention heads:

$$
\mathrm{T}_{\mathrm{bind}}(\boldsymbol{f}, \boldsymbol{H}, \boldsymbol{r}_n) := \quad \underline{\phantom{x}}\!\!\overset{\boldsymbol{r}_n}{\bigcirc} \quad \overset{\boldsymbol{f}}{\bigcirc}\!\!-\!\!\overset{\boldsymbol{H}}{\bigcirc}\!\!\underline{\phantom{x}} \quad = \; \boldsymbol{r}_n \otimes (\boldsymbol{f}^{\top} \boldsymbol{H}).
$$

$$
\mathrm{T}_{\mathrm{bind}}\big(\mathrm{E}_{\mathrm{prop}}(\mathrm{M}_{\mathrm{obj}}(\mathbf{M}, \boldsymbol{r}_m, \boldsymbol{f}_m), \boldsymbol{r}_t\big), \boldsymbol{H}, \boldsymbol{r}_n) = \underline{\phantom{x}}\!\!\overset{\boldsymbol{r}_n}{\bigcirc} \; \sum_s \; \overset{\boldsymbol{f}_m^s}{\bigcirc}\!\!-\!\!\overset{\boldsymbol{f}_m}{\bigcirc} \quad \overset{\boldsymbol{f}_t^s}{\bigcirc}\!\!-\!\!\overset{\boldsymbol{H}}{\bigcirc}\!\!\underline{\phantom{x}}
$$

## C Generalization of TPR-Attention

In previous sections, we have described a TPR-Attention mechanism which matches objects of form order-2 TPRs given a single role-filler query to match. It might often be the case that (i) structured reasoning may require the querying of multiple factors simultaneously, and (ii) objects will need to be represented as a higher–order TPR (see example in Appendix D).

To accommodate for these cases, we first define the general TPR-SAM. Given objects represented by order-$s$ TPRs, $m$ queries and $n$ desired dimensions on our matched objects, we define memory $\mathbf{M}$:

$$
\mathbf{M} = \sum_t \left( \overset{\mathbf{O}_t}{\bigcirc}\!\!\boxed{\vdots} \right\} s \;\Bigg)^{\otimes j} = \sum_t \mathbf{O}_t^{\otimes j},
$$

where $j \cdot s - m = n$ must be satisfied, so that during matching we have

$$
\mathrm{M}_{\mathrm{obj}}\left( \mathbf{M}, \; \mathbf{Q} = \bigotimes_{k=1}^{m} \boldsymbol{q}_k \right) = \overset{\mathbf{Q}}{\bigcirc}\!\!\boxed{\vdots\, m}\,\boxed{\mathbf{M}}\,\boxed{\vdots\, n} \quad .
$$

As an example, consider objects represented as order-2 TPRs:

$$
\mathbf{O}_t \; = \; \sum_i \big( \boldsymbol{r}_i \otimes \boldsymbol{f}_i \big).
$$

If we were to query the simultaneous conjunction of two properties $\boldsymbol{r}_m^1, \boldsymbol{f}_m^1$ and $\boldsymbol{r}_m^2, \boldsymbol{f}_m^2$, we would need the memory to be

$$
\mathbf{M} \; = \; \sum_t \mathbf{O}_t \otimes \mathbf{O}_t \otimes \mathbf{O}_t,
$$

so that we can match the objects having both properties simultaneously:

$$
\mathrm{M}_{\mathrm{obj}}\big( \mathbf{M}, \, (\boldsymbol{r}_m^1, \boldsymbol{f}_m^1), \, (\boldsymbol{r}_m^2, \boldsymbol{f}_m^2) \big) = \begin{matrix} \overset{\boldsymbol{r}_m^1}{\bigcirc} \\ \overset{\boldsymbol{r}_m^2}{\bigcirc} \end{matrix}\!\boxed{\mathbf{M}}\!\begin{matrix} \overset{\boldsymbol{f}_m^1}{\bigcirc} \\ \overset{\boldsymbol{f}_m^2}{\bigcirc} \end{matrix} = \sum_t \begin{matrix} \overset{\mathbf{O}_t}{\bigcirc} \\ \overset{\boldsymbol{r}_m^1}{\bigcirc}\!-\!\overset{\mathbf{O}_t}{\bigcirc}\!-\!\overset{\boldsymbol{f}_m^1}{\bigcirc} \\ \overset{\boldsymbol{r}_m^2}{\bigcirc}\!-\!\overset{\mathbf{O}_t}{\bigcirc}\!-\!\overset{\boldsymbol{f}_m^2}{\bigcirc} \end{matrix} \quad .
$$

## D Action Conditioned TPR-Attention Module

For the composition task described in Section 4, the goal is to construct an output object $\mathbf{O}_3$, whose role-filler structure selectively combines concepts from two input objects. Given

a reference object $\mathbf{O}_1$, a transform object $\mathbf{O}_2$, and an action specifying a target role, the output object should preserve all role-filler bindings from $\mathbf{O}_1$ except for the target role, which is substituted with the corresponding binding from $\mathbf{O}_2$.

Following the general TPR-SAM (as described in Appendix C), we extend the general TPR-Attention baseline (as described in Appendix B), by (i) adding an ID tag to each object to make them identifiable, (ii) changing the query structure to accommodate for the added IDs and specific task, and (iii) constructing queries from the given action.

Given order-3 tensors $\mathbf{O}_t$, two queries (for role and ID, respectively), and a desired single filler dimension on the retrieved objects, we define the memory $\mathbf{M}$ of our task to be the sum of objects $\mathbf{O}_t$,

$$\mathbf{M} = \sum_t \quad \text{—}\left(\mathbf{O}_t\right)\text{—}$$

where each $\mathbf{O}_t$ represents an image latent as a TPR along with an ID tag bound to it with the outer product:

$$\mathbf{O}_t = \sum_j \quad \overset{id^i}{\text{—}\left(r_j\right)\left(f_j^i\right)\text{—}} \quad .$$

We express the desired output object $\mathbf{O}_3$ as a superposition of three components: the contribution from $\mathbf{O}_1$, the removal of the target role-filler binding from $\mathbf{O}_1$, and the contribution from $\mathbf{O}_2$ restricted to the target role.

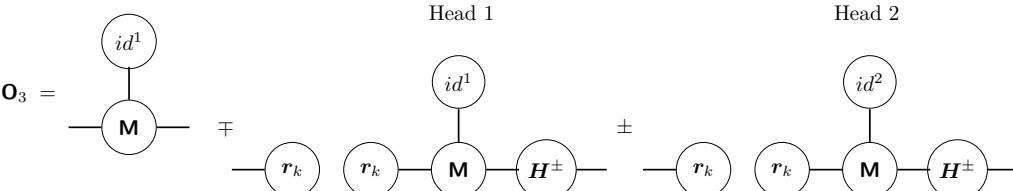

We interpret the composition operation for constructing $\mathbf{O}_3$ as a substitution in the tensor space. Conditioning on $\mathbf{M}$ via $\mathbf{O} \otimes id^1$, we learn to subtract the role-filler corresponding to our chosen action with learned negative identity matrix $\mathbf{H}$. Furthermore, we "substitute" this role-filler with the pair extracted from $\mathbf{O}_2$.

Having defined our desired composition operation for tasks using an action, we now generalize to formalize the learnable parameters of our TPR-Attention mechanism for each head $i$.

We define the query vector $\mathbf{q}_i$ as a function of the input action $\mathbf{a}$. Let $\mathbf{a}$ be the chosen input action and $\mathbf{H}_l^q$ be a learned projection matrix. Then:

$$\mathbf{q}_i(\mathbf{a}) = \left(\mathbf{a}\right)\text{—}\left(\mathbf{H}_l^q\right)\text{—} \quad .$$

Similarly, we define a role $\mathbf{r}_n^i$, which is the role to which our queried filler is bound to in the output object.

$$\mathbf{r}_n^i(\mathbf{a}) = \left(\mathbf{a}\right)\text{—}\left(\mathbf{H}_l^r\right)\text{—} \quad .$$

Finally, we define the generalized attention mechanism at each head as:

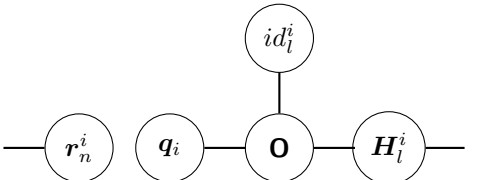

.

## E  COMPOSITION TASK REPRESENTATION DETAILS

We construct an object representation from the ground-truth latent factors of dSprites. Each object is represented as a set of role-filler pairs. Roles are fixed and represented as one-hot vectors. The construction of fillers, on the other hand, depends on the factor of variation.

### E.1  ROLES.

Let $d_r$ denote the number of roles and $d_f$ the filler dimension. We use the canonical basis in $\mathbb{R}^{d_r}$:

$$\boldsymbol{r}_j = \mathbf{e}_j \in \mathbb{R}^{d_r}, \qquad j \in \{1, \ldots, d_r\},$$

so roles are one-hot vectors.

### E.2  NON-INTERACTING FILLERS.

**Color and shape.**  For categorical factors (color, shape), we use one-hot encodings. For example, the three colors red, green, and blue are represented as

$$\boldsymbol{f}_{\text{red}} = \begin{bmatrix} 1 \\ 0 \\ 0 \end{bmatrix}, \quad \boldsymbol{f}_{\text{green}} = \begin{bmatrix} 0 \\ 1 \\ 0 \end{bmatrix}, \quad \boldsymbol{f}_{\text{blue}} = \begin{bmatrix} 0 \\ 0 \\ 1 \end{bmatrix},$$

and shape categories are encoded similarly.

**Orientation.**  Let $\theta \in \mathbb{R}$ denote the orientation angle. We represent orientation on the unit circle:

$$\boldsymbol{f}_{\text{orient}}(\theta) = \begin{bmatrix} \cos \theta \\ \sin \theta \\ 0 \end{bmatrix} \in \mathbb{R}^3.$$

**Position.**  Let $(x, y)$ denote the normalized 2D position. We encode position with:

$$\boldsymbol{f}_{\text{pos}}(x, y) = \begin{bmatrix} x \\ y \\ 1 - \sqrt{x^2 + y^2} \end{bmatrix} \in \mathbb{R}^3.$$

**Scale.**  We map each scale index to a point on the unit sphere using fixed angles $\phi$ and $\theta_s$:

$$\boldsymbol{f}_{\text{scale}}(s) = \begin{bmatrix} \cos \theta_s \\ \sin \theta_s \cos \phi \\ \sin \theta_s \sin \phi \end{bmatrix} \in \mathbb{R}^3,$$

where $\phi$ is fixed and $\{\theta_s\}_{s=0}^5$ are fixed values.

### E.3  INTERACTING FILLERS.

**Numerical interaction.**  Scale and position factors are used to construct an interacting factor, where:

$$\boldsymbol{f}_{\text{scale-pos}} = \text{normalize}(\boldsymbol{f}_{\text{scale}} + \boldsymbol{f}_{\text{pos}})$$

**Categorical interaction.** Shape and color factors are used to construct an interacting factor between two discrete factors, where:

$$\boldsymbol{f}_{\text{shape-color}}[k] = \sum_{i,j} \boldsymbol{f}_{\text{shape}}[i] \; \boldsymbol{M}[k,i,j] \; \boldsymbol{f}_{\text{color}}[j], \qquad k \in \{1, \ldots, d_f\}.$$

# F    Additional Results

## F.1    Extended Results for Non-Interacting Setting

The following results show the loss on combinatorial generalization given 4 heads (left column) and 8 heads (right column) for TPR-Attention and classical attention.

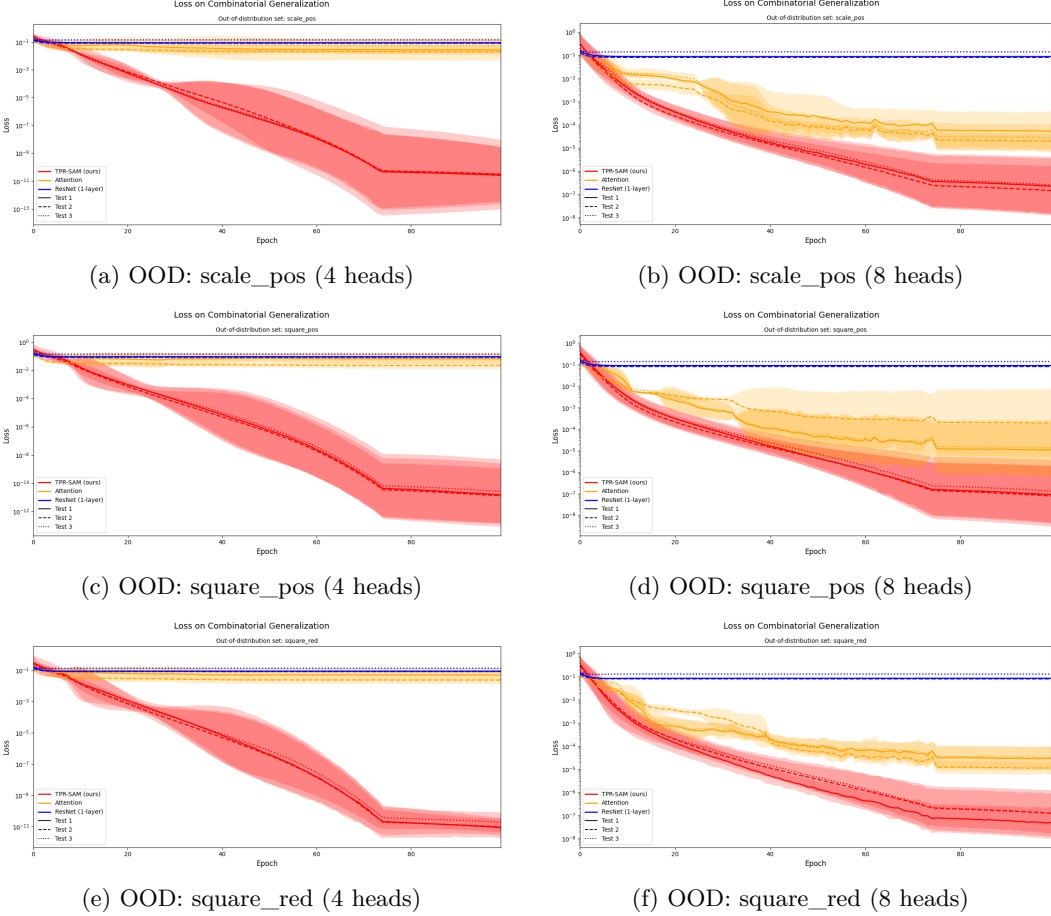

Figure 5: Loss on combinatorial generalization across all out-of-distribution test sets. Left column corresponds to results with 4 heads and the right column to 8 heads. TPR-Attention (red) consistently achieves lower loss than classical attention (yellow) and ResNet baselines (blue). Curves show the mean over 5 seeds with shaded regions indicating $\pm$ 2 standard deviation (4 heads) and $\pm$ 2 standard error (8 heads).

## F.2   Extended Results for Numerical Interaction Setting

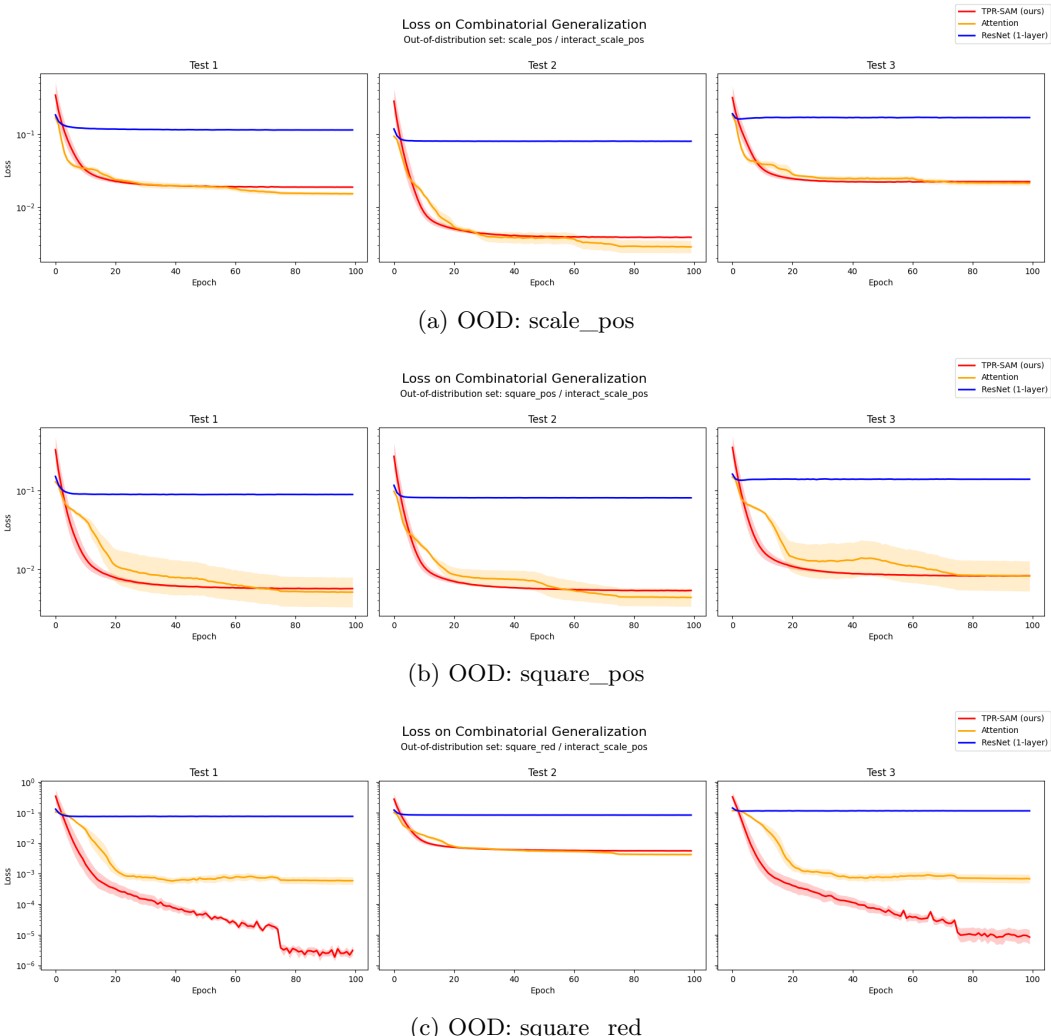

(a) OOD: scale_pos

(b) OOD: square_pos

(c) OOD: square_red

Figure 6: Loss on combinatorial generalization with `scale_pos` interaction across 5 seeds. TPR-Attention (8 heads) shown in red, classical attention (8 heads) in yellow and ResNet baseline with 1 layer in blue. Shaded regions show ± 1 standard error.

## F.3   Extended Results for Categorical Interaction Setting

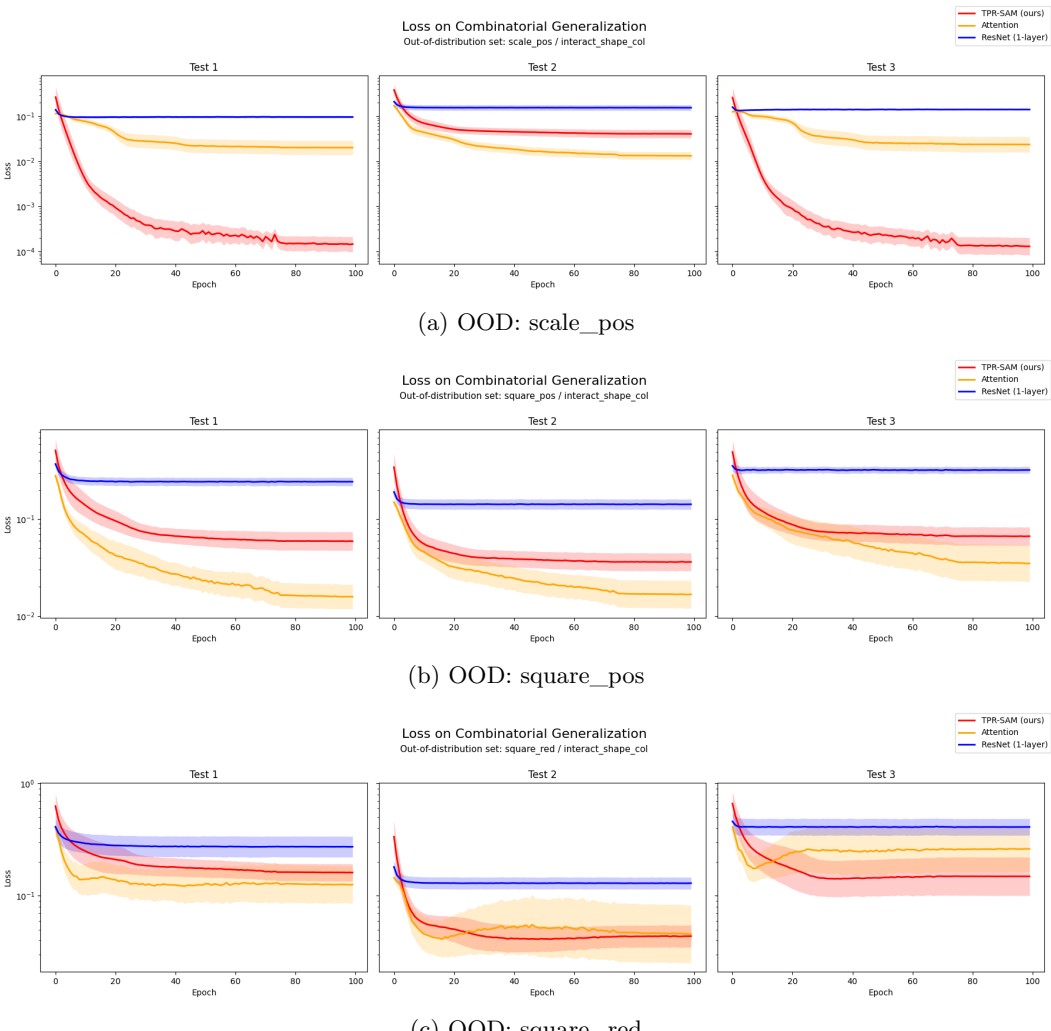

(a) OOD: scale_pos

(b) OOD: square_pos

(c) OOD: square_red

Figure 7: Loss on combinatorial generalization with `shape_col` interaction across 5 seeds. TPR-Attention (8 heads) shown in red, classical attention (8 heads) in yellow and ResNet baseline with 1 layer in blue. Shaded regions show ± 1 standard error.

