# OpenReview forum: "TPR-Attention for Combinatorial Generalization"
_ICLR.cc/2026/Workshop/GRaM — ICLR 2026 Workshop GRaM Poster_

### Official Review · Reviewer_A4qt · 2026-02-09
**Proof-of-concept with manual latents - needs end-to-end learning and multi-layer evaluation**

**Rating:** 4
**Confidence:** 4

**Review:**

## Strengths
* **Novel architecture**: Elegant integration of TPR binding/unbinding operations within attention mechanism
* **Clear presentation**: Tensor network diagrams effectively communicate the approach
* **Systematic evaluation**: Multiple OOD splits and interaction types (numerical, categorical)
* **Tackles hard problem**: Addresses interacting factors where prior disentanglement methods fail
* **Consistent gains**: Shows advantages across different test conditions

## Critical Weaknesses
* **Manual latent provision**: Entire evaluation uses ground-truth dSprites latents—method cannot be applied without separately solving representation learning. Authors acknowledge this but provide no evidence that end-to-end learning is feasible
* **Single-layer comparison only**: Authors explicitly state "no guarantee that the generalization advantage will persist when we compare stacked versions." This caveat invalidates the architectural contribution—single layers insufficient
* **Extremely narrow scope**: Only dSprites with manual TPRs. No other datasets, no real-world tasks
* **Missing compositional baselines**: No comparison with slot attention, relational networks, object-centric methods, or other compositional architectures
* **Potentially unfair**: TPR-Attention has privileged access to ground-truth role-filler structure that baselines don't receive
* **No ablations**: Which components drive improvements? Tensor products? Attention? Structural information?

## Moderate Issues
* **Scalability unclear**: How does this handle many factors, high dimensions, noisy assignments?
* **Limited analysis**: Why does it work? When does it fail? Insufficient mechanistic understanding
* **Implementation details sparse**: Number of heads, dimensions, training setup underspecified
* **No significance tests**: Error bars shown but no statistical validation
* **Generalization unknown**: Would this work on CLEVR, bAbI, SCAN, or other compositional benchmarks?

## Missing Experiments
* End-to-end learning from pixels (critical—even if performance drops, shows feasibility)
* Multi-layer networks (2-4 layers each architecture)
* Slot attention, relational networks, other compositional method baselines
* Ablations isolating binding vs. attention vs. structural information
* Noisy/learned latent factors instead of ground-truth
* Additional compositional datasets beyond dSprites

## Questions
1. Can you show *any* end-to-end learning from pixels, even with lower performance?
2. Do advantages hold with 2-4 stacked layers?
3. How does this compare to slot attention or relational networks?
4. What happens with noisy or learned (not ground-truth) latent factors?
5. Which component drives improvement—tensor products, attention structure, or privileged information?

**Pmlr Suitability:**

NA

---

### Official Review · Reviewer_oXPj · 2026-02-23
**Well-Motived TPR-Attention idea, but Constrained by Artificial Experimental Setup and Weak Baselines**

**Rating:** 3
**Confidence:** 3

**Review:**

**Strengths:**
- Strong framing of the combinatorial generalization problem, which current models struggle with.
- Strong motivation and idea to operate over tensor production representations (TPR) as a way to geometrically build the combinatorial generalization into the architecture.
- Consistent gains across various test conditions considering different OOD splits (scale_pos, square_pos, square_red).

**Weakness:**
- Experiments run entirely on one dataset of hand-crafted latent representations, making it unclear whether gains reflect the attention mechanism itself or the structured inputs. Learning the latent representations end-to-end is a critical and non-trivial piece.
- The baselines are weak only using one layer of standard attention and ResNet not using other object-centric architectural variants.
- Figures contain informal titles, axes, as well as overlapping numbers, axis titles and numbers.
- There are no ablations on where the benefits of using TPRAttention original from in the tested setting.

**Questions:**
- How would the TPRAttention compare without hand-crafted latent representations?
- How would the results scale when multiple layers are used?
- How does the type of OOD matter with numerical (position) vs. categorical (shape color)?

**Pmlr Suitability:**

NA

---

### Official Review · Reviewer_t5ja · 2026-02-24
**Strong idea around Structured Attention via Tensor Product Representations for Compositional Generalization**

**Rating:** 6
**Confidence:** 4

**Review:**

**Overview**

A clear, well-motivated, and conceptually elegant paper shows that structured tensor representations can improve compositional generalization in controlled settings. The proposed mechanism is interesting, interpretable, and potentially broadly applicable. However, the current empirical validation remains limited in scope and scale.

**Weaknesses:**
1) Evaluation restricted to a small, controlled latent setting, which makes it difficult to assess real-world impact or robustness.
2) No experiments on raw perceptual inputs, leaving open whether the method remains effective when representations must be learned rather than provided.
3) Relatively weak baselines.

Despite these limitations, the work is well aligned with the goals of the GRaM workshop, particularly in its emphasis on structured representations, inductive bias, and compositional reasoning. The approach is promising and could have meaningful applications if demonstrated at larger scale.

**Pmlr Suitability:**

NA

---

### Meta-Review · Area_Chair_UrSa · 2026-02-27

**Decision:**

Accept

**Metareview:**

Clever idea to use tensor representations to enable combinatorial generalization, but the experiments remains preliminary.

**Relevance To Proceedings:**

Tiny paper — does not apply

**Relevance To Workshop:**

Yes — suitable for GRaM

---

### Decision · Program_Chairs · 2026-03-02

Accept (Poster)